# The Efficacy and Safety of Immune Checkpoint Inhibitor and Tyrosine Kinase Inhibitor Combination Therapy for Advanced or Metastatic Renal Cell Carcinoma: A Multicenter Retrospective Real-World Cohort Study

**DOI:** 10.3390/cancers15030947

**Published:** 2023-02-02

**Authors:** Koji Iinuma, Toyohiro Yamada, Koji Kameyama, Tomoki Taniguchi, Kei Kawada, Takashi Ishida, Shingo Nagai, Torai Enomoto, Shota Ueda, Kimiaki Takagi, Makoto Kawase, Shinichi Takeuchi, Kota Kawase, Daiki Kato, Manabu Takai, Keita Nakane, Takuya Koie

**Affiliations:** 1Department of Urology, Graduate School of Medicine, Gifu University, Gifu 5011194, Japan; 2Department of Urology, Central Japan International Medical Center, 1-1 Kenkonomachi, Minokamo 5058510, Japan; 3Department of Urology, Ogaki Municipal Hospital, Ogaki 5038502, Japan; 4Department of Urology, Gifu Prefectural General Medical Center, Gifu 5008717, Japan; 5Department of Urology, Gifu Municipal Hospital, Gifu 5008513, Japan; 6Department of Urology, Toyota Memorial Hospital, Toyota 4718513, Japan; 7Department of Urology, Matsunami General Hospital, Hashima-gun 5016062, Japan; 8Department of Urology, Japanese Red Cross Takayama Hospital, 3-113-11 Tenman-machi, Takayama 5068550, Japan; 9Department of Urology, Daiyukai Daiichi Hospital, Ichinomiya 4918551, Japan

**Keywords:** immune checkpoint inhibitor, tyrosine kinase inhibitor, combination first-line treatment, advanced or metastatic renal cell carcinoma, Japanese patients

## Abstract

**Simple Summary:**

We evaluated the efficacy and safety of immune checkpoint inhibitors (ICIs) and tyrosine kinase inhibitors (TKI) (ICI+TKI) in 51 patients with advanced or metastatic renal cell carcinoma (mRCC). In this study, 7.0 months was the median follow-up period. The overall survival rates at 6, 12, and 18 months were 93.1%, 82.5%, and 68.8%, respectively. The median progression-free survival (PFS) was 19.0 months. The objective response and disease control rates were 68.6% and 88.2%, respectively. Patients aged ≥70 years and those with poor-risk mRCC had significantly poorer PFS than their counterparts. ICI+TKI-related adverse events occurred in 43 patients (84.3%) with any grade and in 22 patients (43.1%) with grade ≥3. Treatment selection with poor prognostic factors may be prudent, even though ICI+TKI is an efficacious and safe first-line treatment in patients with mRCC.

**Abstract:**

We evaluated the efficacy and safety of combination therapy with immune checkpoint inhibitors (ICIs) and tyrosine kinase inhibitors (TKI) as first-line therapy for patients diagnosed as having advanced or metastatic renal cell carcinoma (mRCC). We enrolled 51 patients to receive ICI+TKI therapy for mRCC at 9 Japanese institutions. The overall survival (OS) of the patients treated with ICI+TKI was the primary endpoint., and the secondary endpoints were progression-free survival (PFS), objective response rate (ORR), and disease control rate (DCR). Furthermore, we analyzed the clinical prognostic and predictive factors in patients with mRCC treated with ICI+TKI therapy. Seven months was the median follow-up period. The OS rates at 6, 12, and 18 months were 93.1, 82.5, and 68.8%, respectively. The median PFS for patients who received ICI+TKI was 19.0 months, ORR was 68.6%, and DCR was 88.2%. ICI+TKI-related adverse events occurred in 43 patients (84.3%) with any grade and in 22 patients (43.1%) with grade ≥3. Treatment selection with poor prognostic factors may be prudent, even though ICI+TKI is an efficacious and safe first-line treatment in patients with mRCC.

## 1. Introduction

The most common histology of renal cell carcinoma (RCC) is the clear-cell neoplastic component [1,2]. In clear cell RCC, mutations in the von Hippel–Lindau gene, a tumor suppressor, are generally associated with decreased degradation of hypoxia-inducible factors, increased transcription of vascular endothelial growth factor (VEGF), and tumor-induced angiogenesis [3]. Over the past decade, novel tyrosine kinase inhibitors (TKIs) targeting the VEGF receptor, including sunitinib (SUN), axitinib (AXI), and pazopanib, have emerged and have been used to treat advanced and metastatic RCC (mRCC) with completely different effects than previous therapies. Although SUN, AXI, and pazopanib were the standard of care for mRCC, dramatic changes in first-line therapy have occurred with the availability of immune checkpoint inhibitors (ICIs) [4]. Currently, ICIs such as nivolumab (NIVO; programmed cell death protein 1 (PD-1) inhibitor), avelumab (AVE), pembrolizumab (PEM), ipilimumab (IPI; anti-cytotoxic T lymphocyte antigen 4 (CTLA-4) monoclonal antibody), and other ICI (in combination) have become the mainstay of mRCC [5]. The National Comprehensive Cancer Network (NCCN) guidelines recommend combination immunotherapy, including ICI combination therapy or ICI and TKI combination therapy (ICI+TKI), as first-line targeted therapy options for mRCC [6]. The combination of nivolumab and ipilimumab (NIVO+IPI) is the only approved first-line therapy for mRCC in Japan. The CheckMate214 trial was compared the efficacy of NIVO+IPI versus SUN as first-line therapy for patients with clear cell mRCC [7,8]. With a minimum of 5 years of extended follow-up, the international metastatic renal cell carcinoma database consortium (IMDC) risk stratification [9] showed that the median overall survival (OS) of patients with intermediate- or poor-risk mRCC was significantly longer in patients receiving NIVO+IPI than in those receiving SUN (*p* < 0.001) [8]. Furthermore, NIVO+IPI is a treatment modality that has the potential to maintain a durable response for at least 5 years of follow-up in approximately 30% of patients who receive this treatment [8]. As for real-world data, several studies have reported on oncological outcomes and the safety of NIVO+IPI for Japanese patients who diagnosed mRCC [10,11,12,13,14]. The previous trial to assess the benefit and risk associated with NIVO+IPI in patients with mRCC showed OS, progression-free survival (PFS), overall response rate (ORR), and disease control rate (DCR) were similar to those in the CheckMate214 study [10]. On the other hand, there are several ICI+TKI combinations for mRCC, including AVE+AXI, PEM+AXI, NIVO + Cabozantinib (CABO: NIVO+CABO), and PEM + Lenvatinib (LEN: PEM+LEN). Results of randomized phase III trials, including JAVELIN Renal 101 [15], KEYNOTE-426 [16,17], CheckMate 9ER [18,19], and CLEAR [20], reported that I-O+TKI combination therapy demonstrated clinically significant oncologic outcomes compared to SUN or everolimus (EVE). Regarding the outcome of AVE+AXI in patients with mRCC in the United States, an ORR of 50.4% was observed in 365 enrolled patients, and the 1-year OS and PFS were reported to be 39.3% and 73.5%, respectively [21]. However, few studies have evaluated the efficacy and safety of ICI+TKI therapy for mRCC, especially in Japanese patients with mRCC. Therefore, we conducted a multicenter retrospective study to evaluate the efficacy and safety of ICI+TKI in Japanese patients with mRCC and examine the clinical factors that predict disease progression during ICI+TKI treatment.

## 2. Materials and Methods

### 2.1. Patients

This study was authorized by the Institutional Review Board of Gifu University (Approval No.: 2020-271). Since this was a retrospective study, informed consent was not required. Retrospective and observational studies using existing materials and other data have already been published; therefore, written consent was not required in accordance with the Japanese Ethics Committee and Ethical Guidelines. More information on this study is available at https://www.med.gifu-u.ac.jp/visitors/disclosure/docs/2020-271.pdf (accessed on 3 March 2021—in Japanese).

In this retrospective multi-institutional study, patients with mRCC who underwent ICI+TKI therapy at nine Japanese institutions from March 2020 to July 2022 were registered. In this study, mRCC was defined as patients who had simultaneous metastasis at the time of diagnosis of RCC or recurrence after surgery. All enrolled patients were stratified into favorable, intermediate, or poor-risk groups according to the IMDC risk classification [9]. Patients previously treated with TKI or mTOR inhibitor as monotherapy and those with lacking data in the registry were excluded from the entry in this study. The following clinicopathological data were collected in this study: age, sex, evaluation of performance status according to ECOG-PS (Eastern Cooperative Oncology Group, Philadelphia, PA, USA) [22], risk classification using IMDC [9], histological type of primary tumor, levels of serum CRP, surgical history, and therapeutic regimen of ICI+TKI number of metastatic sites, and metastatic sites. Prior to the administration of ICI+TKI, medical history, physical examination, and computed tomography (CT) and/or magnetic resonance imaging (MRI) of the chest, abdomen, and pelvis were evaluated. Additionally, the tumor stage was determined using the American Joint Committee on Cancer Staging Manual tumor staging [23].

### 2.2. Treatment Schedule of ICI+TKI

For AVE+AXI, AVE was administered at a dose of 10 mg/kg body weight every 2 weeks, and AXI was administered orally at 5 mg twice daily. For PEM+AXI, PEM was administered 200 mg intravenously every 3 weeks or 400 mg every 6 weeks, and AXI was administered 5 mg orally twice a day. For NIVO+CABO, NIVO was administered intravenously at 240 mg every 2 weeks or 480 mg every 4 weeks, and CABO was administered orally at 40 mg once daily. In the PEM+LEN group, PEM was administered intravenously at 200 mg every 3 weeks or 400 mg every 6 weeks, and LEN was administered orally at 20 mg once daily. Treatment regimens and doses were determined for each institution, and patients continued treatment until progression of disease as assessed by radiology or intolerable severity of treatment-related adverse events (TRAEs).

### 2.3. Patient Evaluation

For all enrolled patients, CT or MRI was performed every 1–3 months after the initiation of I-O+TKI until the progression of disease by evaluation of radiology or withdrawal of the treatment caused by a TRAEs. According to the response evaluation criteria in solid tumors (RECIST) guidelines (version 1.1) [24], complete response (CR), partial response (PR), stable disease (SD), and progressive disease (PD) were determined and used to evaluate treatment; the best objective response rate (BOR) was used to assess the efficacy of each regimen. ORR was defined as the percentage of patients who obtained a CR or PR following therapy against mRCC; DCR was calculated as the percentage of patients achieving CR, PR, or SD following therapy to mRCC.

### 2.4. Safety

The common terminology criteria for adverse events, ver. 5.0 of the National Cancer Institute [25] was used for evaluating TRAEs and were defined as TRAEs that occurred within at least 100 days of the last administration of ICI+TKI, ICI, or TKI.

### 2.5. Statistical Analysis

OS was assessed as the primary endpoint and PFS, ORR, DCR, and BOR as secondary endpoints. The analysis of the data was carried out using the software JMP 14 (SAS Institute Inc., Cary, NC, USA). The duration of follow-up was determined as the time between the initiation of ICI+TKI administration and the date of the last follow-up check-up or the time of death from any cause. OS was calculated as the interval from the initiation of ICI+TKI administration to the time of death. PFS was determined to be the duration of time from the initiation of ICI+TKI therapy to the date of first confirmed disease progression on imaging studies or death from any cause. The Kaplan–Meier method was used for evaluation of OS and PFS, and the log-rank test was carried out to determine differences due to clinical covariates. Clinical covariates used to predict PFS were evaluated by multivariate analysis using a Cox proportional hazards regression model. The cutoff values for the clinical parameters were determined using a receiver operating characteristic curve analysis [26]. A two-tailed *p*-value < 0.05 was regarded as a statistical significantly different.

## 3. Results

### 3.1. Patients

In the present study, 51 patients with mRCC were registered. The registered patients’ demographic data are listed in Table 1. ECOG-PS was ≥2 in 13 patients (25.5%). Of the three patients who did not have metastasis, one had locally invasive cancer and two had single kidneys with tumors that were difficult to resect by partial nephrectomy. The median duration of follow-up from the initiation of ICI+TKI administration to the analysis date or date of death was 7.0 months (interquartile range (IQR): 4.0–13.0). The number of patients treated with AVE+AXI, PEM+AXI, NIVO+CABO, and PEM+LEN were 19 (37.3%), 19 (37.3%), 8 (15.7%), and 5 (9.7%), respectively.

### 3.2. Efficacy and Oncological Outcomes

The OS rates at 6, 12, and 18 months were 93.1%, 82.5%, and 68.8%, respectively, and the median OS was not reached (95% confidence interval [CI], 18.0-not reached [NR]) (Figure 1a). The PFS was 77.4% at 6 months, 72.8% at 12 months, and 65.5% at 18 months, with a median of 19.0 months (95% CI, 14.0-NR) (Figure 1b).

With respect to age, the median PFS for patients aged ≥70 years was significantly shorter than that for patients aged <70 years (19.0 months vs. NR, *p* = 0.018) (Figure 2a); with respect to IMDC risk stratification, the median PFS was significantly shorter for patients at poor risk than for those at good/intermediate risk (6.0 months vs. NR, *p* = 0.038) (Figure 2b). There was no significant association between PFS and sex (*p* = 0.24), number of metastatic sites (*p* = 0.80), or CRP level (*p* = 0.51).

Table 2 shows the treatment effect in patients with mRCC who received ICI+TKI therapy. In this study, the ORR was observed in 35 patients (68.6%) and DCR in 45 patients (88.2%). Regarding the BOR, 5 patients (9.8%) achieved CR, and 30 (58.8%) achieved PR.

In the multivariate analysis, age ≥70 years and poor risk in the IMDC risk classification were statistically correlated with poor PFS in patients with mRCC treated with ICI+TKI (Table 3).

### 3.3. Safety

TRAEs are presented in Table 4. Six patients (11.8%) required high-dose glucocorticoids (>40 mg prednisone/day or equivalent); no patients received immunosuppressive drugs for the management of TRAEs. Seventeen patients (33.3%) discontinued ICI+TKI treatment owing to TRAEs. In contrast, no TRAE-related deaths occurred among the enrolled patients during the follow-up period.

## 4. Discussion

Currently, the era of monotherapy with ICI or TKI has changed to that of combination therapy. In fact, the number of treatment options for mRCC has increased significantly, and the concept of treatment for mRCC seems to be changing dramatically. Therefore, it is necessary to consider which treatment to choose for which patients and how to formulate a treatment strategy that includes second-line treatment and beyond, making the choice of first-line treatment difficult. Numerous randomized phase III trials have made comparisons of benefit and risk between ICI+TKI and SUN or EVE as the conventional standard of care in patients with mRCC [15,16,17,18,19,20]. Table 5 shows the summary of the phase III trial of ICI+TKI available in Japan.

The JAVELIN RENAL 101 trial randomized 886 enrolled patients with mRCC who received AVE+AXI or SUN and compared PFS and OS as the primary endpoint for patients with PD-L1-positive tumors and PFS as a secondary endpoint for all patients [15]. The median PFS was 13.8 months with AVE+AXI and 7.2 months with SUN (hazard ratio (HR), 0.61; 95% CI, 0.47–0.79; *p* < 0.001) [15]. In contrast, the median PFS for all patients had been 13.8 months using AVE+AXI and 8.4 months using SUN (HR, 0.69; 95% CI, 0.56–0.84; *p* < 0.001) [15]. In the second interim analysis, at a minimum follow-up period of 13 months, PFS was similar to that in the aforementioned study, and OS was not reached [27]. The open-label phase III trial conducted by KEYNOTE-426 enrolled 861 patients with mRCC who received PEM+AXI or SUN with a median follow-up period of 12.8 months [16]. The estimated OS rate at 12 months was 89.9% in patients receiving PEM+AXI and 78.3% in those receiving SUN (HR for death, 0.53; 95% CI, 0.38–0.74; *p* < 0.001) [16]. The median PFS in patients receiving PEM+AXI treatment was 15.1 months and 11.1 months in those receiving SUN (HR, 0.69; 95% CI, 0.57–0.84; *p* < 0.001) [16]. The ORR for patients who received PEM+AXI was significantly higher than that for those who received SUN (59.3% vs. 35.7%; *p* < 0.001) [16]. At a median follow-up period of 30.6 months, the clinical benefit of PEM+AXI continued versus SUN in OS (HR, 0.68; 95% CI, 0.55–0.85; *p* = 0.0003) and PFS (HR, 0.71; 95% CI, 0.60–0.84, *p* < 0.001) [17]. The CheckMate 9ER trial is a phase 3, open-label, randomized study comparing the therapeutic efficacy of NIVO+CABO and SUN in patients with mRCC [19]. In this study of 651 patients with a median follow-up of 18.1 months, the median PFS was 16.6 months with NIVO+CABO and 8.3 months with SUN (HR, 0.51; 95% CI, 0.41–0.64; *p* < 0.001) [18]. Additionally, the one-year OS rates were 85.7% and 75.6% with NIVO+CABO and SUN, respectively (HR, 0.60; 95% CI, 0.40–0.89; *p* = 0.001) [18]. Regarding ORR, patients receiving NIVO+CABO had 55.7% and those receiving SUN had 27.1% (*p* < 0.001) [19]. At an extended median follow-up period of 32.9 months, OS and PFS were significantly prolonged in patients who received NIVO+CABO compared with those who received SUN (*p* = 0.0043 and *p* < 0.0001, respectively) [19]. In the CLEAR trial comparing the efficacy of PEM+LEN, EVE+LEN, and SUN in patients with mRCC, PFS was significantly prolonged in patients who received PEM+LEN (HR, 0.39; 95% CI, 0.32–0.49; *p* < 0.001) and EVE+LEN (HR, 0.65; 95% CI, 0.53–0.80; *p* < 0.001) than in those who received SUN [20]. However, OS was significantly prolonged with PEM+LEN (HR, 0.66; 95% CI, 0.49–0.88; *p* = 0.005) but not with EVE+LEN (HR, 1.15; 95% CI, 0.88–1.50; *p* = 0.30), compared that in patients who received SUN [20]. Therefore, the CLEAR trial concluded that PEM+LEN significantly improved PFS and OS compared with SUN [20]. As for real-world data, Yang et al. reported the efficacy and safety of ICI+TKI in any treatment line for mRCC [28]. In this study, they found that 38.8% of patients received ICI+TKI as first line, with an ORR of 56.7% and a median PFS of 15.2 months [28]. In our study, the OS, PFS, ORR, and DCR of patients treated with various combinations of ICI+TKI were similar to those in some of the aforementioned phase III studies. We also reported the efficacy and safety of NIVO+IPI for mRCC [10,11]. Namely, the ORR in moderate- and poor-risk patients based on the IMDC risk classification ranged from 34.3 to 41.9%, with OS and PFS at 6 months ranging from 94.7 to 100% and 76.1 to 78.6%, respectively, and OS and PFS at 12 months ranging from 89.1 to 95.8% and 56.2 to 63.1%, respectively [10,11]. However, it is difficult to simply compare these studies because of the differences in patient background, time of investigation, and follow-up period. On the other hand, based on these results, it may be safe to state that OS and PFS are almost equivalent, although the ORR may be better with ICI+TKI than with NIVO+IPI. On the other hand, ICI+TKI treatment did not show a durable response as seen with NIVO+IPI, and it remains unclear which treatment is more effective for patients with mRCC. Therefore, it is necessary to investigate the clinical and pathological factors, including biomarkers, that might predict some treatment effects.

This study suggests that age may be a prognostic factor for PFS, whereas there have been few studies on age and treatment effects for mRCC. Tomita et al. [29] reported that AVE+AXI showed better efficacy than SUN in oncological outcomes, such as OS and PFS, in all age groups, including patients aged ≥75 years. The HRs of both PFS and OS appeared to be higher in patients aged 65–74 and ≥75 years than in those aged ≤64 years [29]. Patients over 75 years of age who underwent AVE+AXI treatment would have a similar survival benefit as those aged 65–74 years [29]. A meta-analysis of the literature on the combination of ICI+TKI in the management of older patients with mRCC showed that PEM+AXI prolonged PFS compared to SUN in patients with age ≥65 years (HR, 0.63; 95% CI, 0.45–0.88) [30]. Similarly, AVE+AXI led to a prolonged PFS compared to SUN (HR, 0.70; 95% CI, 0.49–0.99) [30]. A recent update on the efficacy of the JAVELIN RENAL 101 trial showed no difference in median PFS or OS between patients aged 65–74 and ≥75 years [27]. Eggers and colleagues [31] stated that although advancing age may be an adverse predictor of survival, it may not be a reason to withhold TKIs or all new treatment options that may be recommended in the future. Indeed, a pooled analysis of phase II and III trials on mRCC in 4736 patients examined whether age was an independent prognostic predictor of OS [32]. When stratified into three groups, young (<50 years), middle-aged (50–70 years), and elderly (>70 years), the OS was 20, 17.3, and 21 months, respectively, with no significant differences among the three groups [32]. For a variety of malignant neoplasms, ICIs did not provide a survival benefit compared to standard therapy in pretreated patients aged ≥75 years (HR, 0.96; 95% CI, 0.72–1.26; *p* = 0.39) [33]. Although age ≥75 years had been an independent risk factor for mortality, it is currently reported that age is not a risk factor for mortality [30]; this is an issue that should be considered in the future.

The IMDC risk classification was originally developed as a model for predicting mRCC prognosis in the era of TKI therapy, and its classification showed a clear difference in prognosis in patients treated with TKI therapy for mRCC [9]. Even in the ICI era, the IMDC risk classification is still used for treatment selection for mRCC, and there are several reports on the validity and usefulness of the IMDC risk classification of NIVO+IPI for mRCC and its problems [10,34]. Multivariate analysis showed a correlation between treatment and the number of IMDC risk factors, which were significant for OS (*p* = 0.043), PFS (*p* = 0.003), and ORR (*p* < 0.001) [34]. At a minimum 30-month follow-up, the ORR for NIVO+IPI was consistent with a risk factor of zero to six and comparable to the proportion of intent-to-treat (ITT) patients [34]. Similarly, CR rates by NIVO+IPI were higher than those by SUN for zero to three risk factors [34]. No CR was observed in either group with four or six risk factors [34]. With respect to oncologic outcomes, OS and PFS of patients treated with NVO+IPI were significantly longer than those of patients treated with SUN across one–six IMDC risk factors [34]. In a multivariate analysis of individual IMDC risk factors in CheckMate214, it was found that in the NIVO+IPI group, four of the six factors (anemia, neutrophilia, thrombocytopenia, time from diagnosis) were not prognostic for OS, whereas two factors (Karnofsky Performance Status, corrected calcium) were prognostic factors, which is noteworthy [34]. Likewise, the JAVELIN Renal 101 and KEYNOTE-426 trials using ICI+TKI showed significantly worse OS, PFS, and ORR as the number of IMDC risk factors increased [15,16]. The study concluded that a consistent efficacy of NIVO+IPI compared to SUN was observed in all intermediate-/low-risk patients with mRCC in the CheckMate214 trial, regardless of the number of individual risk factors in the IMDC risk classification [35]. Certainly, the CR rate is the most desirable outcome, albeit not necessarily the only important response outcome; other prognostic parameters besides the IMDC risk category may be important. Powel [36] stated that it is premature and unwise to compare CR rates when the difference between trials is ≤5%, suggesting that one is associated with better outcomes without considering other factors. We have previously demonstrated that in patients treated with NIVO+IPI for mRCC, poor-risk patients based on the IMDC risk classification were significantly associated with poor PFS [10]. Furthermore, in this study, poor-risk patients had significantly poorer PFS than good/intermediate-risk patients. Thus, it remains unclear which therapy is optimal for patients with poor-risk mRCC. Considering the poor PFS in poor-risk patients, as shown in this study, treatment selection should be considered on a case-by-case basis. Therefore, it may be necessary to reconsider the argument that NIVO+IPI should not be chosen for patients with a high number of IMDC risk factors because of the lack of CR.

Regarding safety, 22 patients (43.1%) experienced grade 3/4 TRAEs. Of these, 6 (11.8%) received high-dose glucocorticoids and 17 (33.3%) discontinued ICI+TKI treatment because of TRAEs. In the JAVELIN RENAL 101 trial, grade ≥3 TRAEs were observed in 71.2% of patients receiving AVE+AXI and 71.5% of those receiving SUN [15]. For both AVE and AXI, 33 (7.6%) discontinuations were due to TRAEs [15]. Among patients receiving AVE+AXI, three patients (0.7%) had TRAE-related deaths (sudden death in one patient, myocarditis in one, and necrotizing pancreatitis in one) [15]. Forty-eight patients (11.1%) treated with AVE+AXI received high-dose glucocorticoids (prednisone ≥40 mg total daily dose) because of TRAEs [15]. In the KEYNOTE 426 trial, TRAEs occurred in 75.8% of patients receiving PEM+AXI and in 70.6% of those receiving SUN, with a TRAE grade ≥3 [16]. In PEM+AXI treatment, when a TRAE occurred for any reason, 30.5% of patients discontinued either of the drugs, 10.7% discontinued both drugs, 69.9% interrupted each of the drugs, and 20.3% reduced the AXI dose [16]. According to the CheckMate 9ER trial, grade ≥3 TRAEs of any cause were observed in 75.3% of patients treated with NIVO+CABO and in 70.6% of those treated with SUN [18]. A total of 19.7% of patients who underwent NIVO+CABO discontinued the study agents because of any TRAEs, including 6.6% of patients who discontinued NIVO only, 7.5% who discontinued CABO only, and 5.6% who discontinued both NIVO and CABO [18]. Seventy patients (22%) treated with NIVO+CABO were administered corticosteroids (prednisone ≥ 40 mg/day) for any duration to manage TREAEs; 40 (13%) received corticosteroids continuously for at least 14 days; and 16 (5%), for at least 30 days [19]. In the CLEAR trial, grade ≥3 TRAEs, including diarrhea, hypertension, elevated lipase levels, and hypertriglyceridemia, developed in 82.4% of patients receiving PEM+LEN and in 71.8% of those receiving SUN [20]. Among patients treated with PEM+LEN, 37.2% had a TRAE of any grade and discontinued PEM, LEN, or both drugs (PEM, 28.7%; LEN, 25.6%; both, 13.4%); 68.8% reduced the dose of LEN; and 78.4% discontinued PEM, LEN, or both drugs [20]. In this study, the incidence of grade 3/4 TRAEs, the proportion of patients who received high-dose glucocorticoids for TRAEs, and the proportion of patients who discontinued ICI+TKI treatment due to TRAEs were similar to the results of the aforementioned phase III trials. In daily practice, a certain percentage of patients may develop TRAEs caused by ICIs and TKIs, even though ICI+TKI therapy for patients with mRCC seems to be relatively safe. Therefore, it is very important to accurately identify which drug was responsible for developing TRAEs and to administer the appropriate treatment according to the symptoms, at the appropriate time.

In phase III studies including JAVELIN RENAL 101, KEYNOTE 426, CheckMate 9ER, and CLEAR studies, ≥grade 3 increases in alanine aminotransferase (ALT), aspartate aminotransferase (AST), ICI +TKI were observed in 3.1–7.0% and 3.4–13.3% of patients treated with TKIs [15,16,17,18,19,20,27]. In a subgroup analysis of Japanese patients in the JAVELIN RENAL 101 study reported by Uemura et al. [37], grade ≥3 ALT and AST elevations were found in 6.0% and 3.9% of the total population in the AVE+AXI group, compared with the incidence tended to be higher in the Japanese population, 12.1% and 6.0%, respectively, in the AVE+AXI group. According to a subgroup analysis of Japanese patients in the KEYNOTE 426 study, 11.4% of Japanese patients in the PEM+AXI group had liver dysfunction of grade 3 or higher [38]. In the present study, the incidence of grade ≥3 hepatobiliary system disorders were 13.7%, which is comparable to the results of the subgroup analysis of Japanese patients. A higher incidence of TRAEs has been reported in Japanese or Asian patients with RCC who received VEGF/VEGF receptor inhibitors compared to non-Asian patients [39,40,41,42]. Therefore, it is possible that racial differences in TRAEs during treatment with ICI+TKIs may be observed, especially in Japanese patients, who may have more hepatobiliary diseases, and therefore, more careful follow-up may be necessary when treating mRCC with ICI+TKIs.

This study has several limitations. First, because this is a retrospective, multicenter, single-arm study of data from patients with mRCC treated with ICI+TKI, there may be variations in the diagnosis and treatment between centers, which has an inherent potential for bias. Furthermore, the number of ICI+TKI treatment modalities for mRCC has increased over time, and the real-world clinical situation is changing in terms of the treatment modality chosen. Second, because of the relatively small number of the enrolled patients and the relatively short follow-up duration, it seemed necessary to be careful in interpreting the results, especially with regard to PFS. Third, clear cell, papillary, and unclassifiable RCCs were targeted, and an additional 11.8% had an unknown histologic type. Therefore, the possibility that the lack of a uniform histological type may have affected the oncological outcomes should be considered. In the future, it may be necessary to examine the oncological outcome of different histologic types. Fourth, since this is an observational study, translational research is lacking, and this seems to be an issue for the future.

## 5. Conclusions

We have retrospectively evaluated in patients with mRCC of the efficacy and safety associated with regimens of ICI+TKI. According to the results of this study, the oncologic outcomes and safety of ICI+TKI were comparable to those of previous trials. Thus, management of ICI+TKI potentially may have advantages as a useful first-line treatment option in patients with mRCC, leading to positive treatment effects. Furthermore, age ≥70 years and poor risk based on the IMDC risk classification might be associated with poor PFS in patients with mRCC treated with ICI+TKI. Treatment selection with poor prognostic factors may be prudent, even though ICI+TKI is an efficacious and safe first-line treatment in patients with mRCC. More research are required with a larger number of patients and with long-term follow-up.

## Figures and Tables

**Figure 1 cancers-15-00947-f001:**
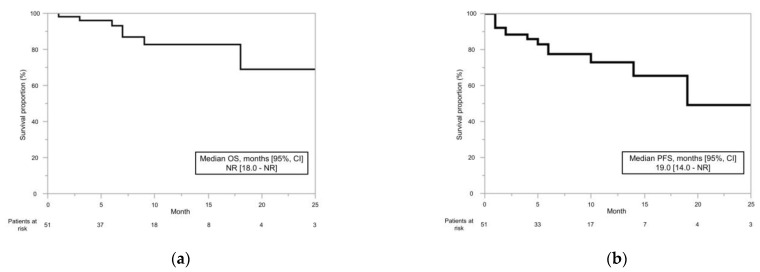
Overall survival (OS) and progression-free survival (PFS) in patients with advanced or metastatic renal cell carcinoma who received immune checkpoint inhibitors and tyrosine kinase inhibitors were estimated using the Kaplan–Meier method. (**a**) OS at 6, 12, and 18 months was 93.1%, 82.5%, and 68.8%, respectively, and did not reach the median OS. (**b**) PFS at 6, 12, and 18 months was 77.4%, 72.8%, and 65.5%, respectively, and the median PFS was 19.0 months.

**Figure 2 cancers-15-00947-f002:**
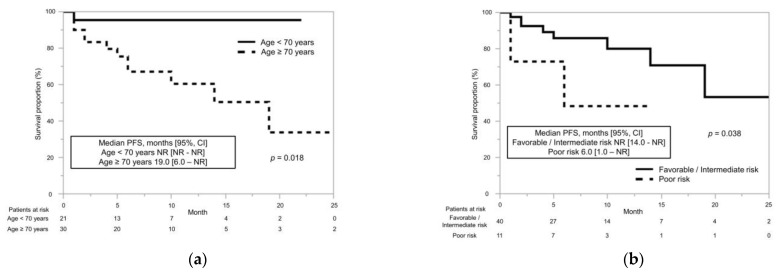
Progression-free survival (PFS) in patients with advanced or metastatic renal cell carcinoma (mRCC) who were administered immune checkpoint inhibitors and tyrosine kinase inhibitors (ICI+TKI) was estimated using the Kaplan–Meier method. (**a**) Patients with mRCC treated with ICI+TKI were divided by age, and their PFS was compared using the Kaplan–Meier method. The median PFS in patients with age ≥70 years was significantly shorter than that in patients with age <70 years (19.0 months vs. not reached (NR), *p* = 0.018) (**b**) Patients with mRCC treated with ICI+TKI were divided by the International Metastatic Renal Cell Carcinoma Database Consortium risk classification. The median PFS was significantly shorter in patients with poor risk than in those with favorable/intermediate risk (6.0 months vs. NR, *p* = 0.038).

**Table 1 cancers-15-00947-t001:** Characteristics of the enrolled patients.

Covariates	
Age (year, median, interquartile range)	71 (61–75)
Sex (number, %)	
Male	36 (70.6)
Female	15 (29.4)
ECOG-PS (number, %)	
0	26 (51.0)
1	12 (23.5)
2	6 (11.8)
3	6 (11.8)
4	1 (1.9)
Primary IMDC risk classification (number, %)	
Favorable risk	12 (23.5)
Intermediate risk	28 (54.9)
Poor risk	11 (21.6)
Histology of primary lesions (number, %)	
Clear cell renal cell carcinoma	36 (70.6)
Papillary renal cell carcinoma	4 (7.8)
Unclassified	5 (9.8)
Unknown	6 (11.8)
CRP (mg/dL, median, interquartile range)	0.33 (0.07–2.31)
Patients who underwent nephrectomy before treatment (number, %)	32 (62.7)
Number of metastatic sites (number, %)	
0	3 (5.9)
1	30 (58.9)
2	12 (23.5)
3	4 (7.8)
4	2 (3.9)
Total number of metastatic sites (number, %)	
Lung	29 (56.9)
Bone	12 (23.5)
Lymph node	11 (21.6)
Adrenal gland	4 (7.8)
Pancreas	3 (5.9)
Brain	2 (3.9)
Others	6 (11.8)

ECOG-PS: Eastern Cooperative Oncology Group performance status; IMDC: International Metastatic Renal Cell Carcinoma Database Consortium; CRP: C-reactive protein; ICI+TKI: immune checkpoint inhibitor and tyrosine kinase inhibitor.

**Table 2 cancers-15-00947-t002:** The treatment effect in patients who received immune checkpoint inhibitor and tyrosine kinase inhibitor.

	AVE+AXI (*n* = 19)	PEM+AXI (*n* = 19)	NIVO+CABO (*n* = 8)	PEM+LEN (*n* = 5)
ORR (number, %)	10 (52.6)	16 (84.2)	6 (75.0)	3 (60.0)
DCR (number, %)	15 (79.0)	18 (94.7)	8 (100)	4 (80.0)
BOR (number, %)				
CR	2 (10.5)	3 (15.8)	0	0
PR	8 (42.1)	13 (68.4)	6 (75.0)	3 (60.0)
SD	5 (26.3)	2 (10.5)	2 (25.0)	1 (20.0)
PD	4 (21.1)	1 (5.3)	0	1 (20.0)

AVE+AXI: avelumab + axitinib; PEM+AXI: pembrolizumab + axitinib; NIVO+CABO: nivolumab + cabozantinib; PEM+LEN: pembrolizumab + lenvatinib; ORR: objective response rate; DCR: disease control rate; BOR: best objective response; CR: complete response; *n*: number; PD: progressive disease; PR: partial response; SD: stable disease.

**Table 3 cancers-15-00947-t003:** Multivariate analysis of clinical parameters for the prediction of progression-free survival.

	*n*	Multivariate Analysis		
		HR	95% CI	*p*
Age				
<70	21	1 (ref.)	-	-
≥70	30	8.23	1.01–66.8	0.048
Gender				
Male	36	1 (ref.)	-	-
Female	15	0.34	0.07–1.57	0.70
IMDC risk classification				
Favorable/intermediate risk	40	1 (ref.)	-	-
Poor risk	11	19.9	2.75–143.8	0.003
Number of metastatic sites				
≤1	33	1 (ref.)	-	-
≥2	18	0.08	0.22–3.49	0.86
CRP				
<1.25 mg/dL	35	1 (ref.)	-	-
≥1.25 mg/dL	16	0.10	0.01–1.02	0.052

*n*: number; HR: hazard ratio; CI: confidence interval; ref.: reference; IMDC: International Metastatic Renal Cell Carcinoma Database Consortium; CRP: C-reactive protein.

**Table 4 cancers-15-00947-t004:** Treatment-related adverse events of immune checkpoint inhibitors and tyrosine kinase inhibitors.

Event (Number, %)	All Grades	Grades 3/4
Any events	43 (84.3)	22 (43.1)
Hypothyroidism	12 (23.5)	1 (2.0)
Palmar–plantar erythrodysesthesia syndrome	9 (17.6)	3 (5.9)
Diarrhea	8 (15.7)	1 (2.0)
Hepatobiliary disorders	8 (15.7)	7 (13.7)
Hypertension	7 (13.7)	2 (3.9)
Proteinuria	7 (13.7)	2 (3.9)
Colitis	5 (9.8)	2 (3.9)
Adrenal insufficiency	4 (7.8)	3 (5.9)
Malaise	3 (5.9)	0
Anorexia	3 (5.9)	0
Erythema multiforme	2 (3.9)	1(2.0)
Myositis	2 (3.9)	1 (2.0)
Hyperthyroidism	2 (3.9)	0
Others	7 (13.7)	5 (9.8)

**Table 5 cancers-15-00947-t005:** Summary of a phase III study of combination therapy with immune checkpoint inhibitors and tyrosine kinase inhibitors for metastatic renal cell carcinoma.

	JAVELIN Renal 101 [15,27]		KEYNOTE-426 [16,17]		CheckMate 9ER [18,19]		CLEAR [20]		
	AVE+AXI	SUN	PEM+AXI	SUN	NIVO+CABO	SUN	PEM+LEN	EVE+LEN	SUN
number	442	444	432	429	323	328	355	357	357
IMDC risk classification (%)									
Favorable risk	21	23	32	31	23	22	31	32	35
Intermediate risk	64	66	55	57	58	57	59	55	54
Poor risk	12	10	13	12	19	21	9	12	10
Median follow- up (months)	19.3		30.6		32.9		26.6		
Median OS (months)	NR	NR	NR	35.7	37.7	34.3	NR	NR	NR
Median PFS (months)	13.3	8.0	15.4	11.1	16.6	8.3	23.9	14.7	9.2
ORR (%)	53	27	60	40	56	28	71	54	36
CR (%)	4	2	9	3	12	5	16	10	4
PR (%)	49	25	51	37	43	23	55	44	32
SD (%)	28	44	23	35	33	41	19	34	38
PD (%)	12	19	11	17	6	14	5	7	14
≥Grade 3 TRAE (%)	71	72	67	62	65	54	82	83	72

AVE+AXI: avelumab + axitinib; PEM+AXI: pembrolizumab + axitinib; NIVO+CABO: nivolumab + cabozantinib; PEM+LEN: pembrolizumab + lenvatinib; EVE+LEN: evelorimus + lenvatinib; IMDC: International Metastatic Renal Cell Carcinoma Database Consortium; OS: overall survival; PFS: progression-free survival; ORR: objective response rate; CR: complete response; PD: progressive disease; PR: partial response; SD: stable disease; TRAE: treatment-related adverse events.

## Data Availability

Data and materials are provided in this paper.

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
