# Peer review of "The Efficacy and Safety of Immune Checkpoint Inhibitor and Tyrosine Kinase Inhibitor Combination Therapy for Advanced or Metastatic Renal Cell Carcinoma: A Multicenter Retrospective Real-World Cohort Study"

_cancers, 2023, doi:10.3390/cancers15030947_

Round 1

Reviewer 1 Report

The authors present to us a real-world study in the treatment of renal cell carcinoma. Real world studies are an important tool to understand the applicability of clinical trials in practical settings;

The paper present a small sample of patients with metastatic renal cell carcinoma and this can represent a major limitation to the correct interpretation of the study, but the authors try to solve this limitation with a multi centric design.

Introduction:

- good introduction to fundament the importance of the study

- line 64: instead of “… and other ICI have become the mainstay of metastatic RCC” I suggest that it should be “and other ICI (in combination) have become the mainstay of metastatic RCC”.

Material and Methods

 - Explain correctly and accurately the methods

- The details of the ethical approval can be accessed at the site of Gifu University, but is only in japanese and a disclosure should be added, for example (accessed on March 3, 2021 – in Japanese).

 - Minor Error in line 110 – performance “tatus” where it should be status

Results

There are some flaws in the presentation of the results that could be addressed:

-  For analysis all patients with RCC were grouped, despite approximately 30% of the patients didn’t have cc RCC. The authors could do some explanation about this topic

 -  6% of patients had no metts – what was the reason for systemic therapy? I think that this should be explained also (in the results/discussion topic)

- 32% of patiens underwent nephrectomy before treatment – the presentation of metastatic disease was synchronous or metachronous? - Some considerations can be made in order to clarify this

Discussion 

I think that this section deserve some "re-doing", because the authors present to us several results of different trials.

My suggestion should be to address the findings of different trials in comparison with their results - for examples a table of OS for different studies and comparison with population in study should be easier for the reader to apprehend the results.

The limitations should discuss the small number of patients, the different pathology groups should be addressed in comparison with different trials.

Author Response

January 26, 2023

Prof. Dr. Samuel C. Mok

Editor-in-Chief, Cancers

Dear Editor:

Thank you very much for the review of our manuscript titled “The Efficacy and Safety of Immune Checkpoint Inhibitor and Tyrosine Kinase Inhibitor Combination Therapy for Advanced or Metastatic Renal Cell Carcinoma: A Multicenter Retrospective Real-world Cohort Study”.

We sincerely appreciate all valuable comments and suggestions, which helped us to improve the quality of our manuscript. Our responses to the reviewers’ comments are described below in a point-to-point manner. Appropriate changes, suggested by the Reviewers, have been introduced to the manuscript (track-changes mode in the red color font). Let me emphasize our full readiness to make any further improvements to the manuscript.

We hope that our manuscript will be acceptable for publication in the Cancers.

We look forward to hearing from you.

Yours sincerely,

Takuya Koie

Department of Urology

Gifu University Graduate School of Medicine

1-1 Yanagido, Gifu, Gifu 501-1194, Japan

TEL.: +81-582-30-6338

FAX: +81-582-30-6341

Responses to the reviewer's comments

We would like to thank the Reviewers for taking the time and effort necessary to review the manuscript. We sincerely appreciate all the valuable comments and suggestions, which helped us to improve the quality of the manuscript.

Response to Reviewer 1

The authors appreciate the reviewer’s comments. The authors’ point-by-point responses to the comments are given below.

  1. Introduction:

- good introduction to fundament the importance of the study

- line 64: instead of “… and other ICI have become the mainstay of metastatic RCC” I suggest that it should be “and other ICI (in combination) have become the mainstay of metastatic RCC”.

Response:

The authors have revised the sentence on line 68:

and other ICI (in combination) have be-come the mainstay of mRCC and others have become the mainstay of metastatic RCC (mRCC) treatment [5].

  1. Material and Methods

-Explain correctly and accurately the methods

-The details of the ethical approval can be accessed at the site of Gifu University, but is only in Japanese and a disclosure should be added, for example (accessed on March 3, 2021 – in Japanese).

Response:

The authors have revised the sentence on line 108:

(accessed on March 3, 2021 - in Japanese).

  1. Minor Error in line 110 – performance “tatus” where it should be status

Response:

On line 117, The authors have changed from “tatus” to “status.”

  1. Results

There are some flaws in the presentation of the results that could be addressed:

For analysis all patients with RCC were grouped, despite approximately 30% of the patients didn’t have cc RCC. The authors could do some explanation about this topic.

Response:

The authors fully agree with the reviewers' comments. One problem with this study is that it included 30% non-ccRCC patients. It is possible that different histologic types may have different efficacy against treatment, and it may be necessary to evaluate cancer outcomes by histologic type in the future. Therefore, the authors added the following text to line 415:

In the future, it may be necessary to examine the oncological outcome of different histologic types.

  1. 6% of patients had no mets – what was the reason for systemic therapy? I think that this should be explained also (in the results/discussion topic)

Response:

The authors have added the following sentence on line 166:

Of the three patients who did not have metastasis, one had locally invasive cancer and two had single kidneys with tumors that were difficult to resect by partial nephrectomy.

  1. 32% of patients underwent nephrectomy before treatment – the presentation of metastatic disease was synchronous or metachronous? - Some considerations can be made in order to clarify this

Response:

The authors have added the following sentence on line 110:

In this study, mRCC was defined as patients who had simultaneous metastasis at the time of diagnosis of RCC or recurrence after surgery.

  1. I think that this section deserves some "re-doing", because the authors present to us several results of different trials.

My suggestion should be to address the findings of different trials in comparison with their results - for examples a table of OS for different studies and comparison with population in study should be easier for the reader to apprehend the results.

Response:

The authors have added the Table 5 with regard to summary of a phase III study of combination therapy with immune checkpoint inhibitors and tyrosine kinase inhibitors for metastatic RCC on line 236.

Reviewer 2 Report

The authors have investigated the efficacy and safety of combination therapy including immune checkpoint inhibitors (ICI) and tyrosine kinase inhibitors (TKI) for patients with metastatic renal cell carcinoma. The study led to the finding that the combination of ICI+TKI is more effective than the first-line treatment options available for mRCC. 

It is an interesting finding.

However, similar such study by Yang et al; has been already reported in “Combining immune checkpoint inhibition plus tyrosine kinase inhibition as first and subsequent treatments for metastatic renal cell carcinoma” published in Cancer Medicine 11, 2022, 3106.

The authors have not cited this paper. Authors need to discuss this paper in their manuscript and the need to compare the outcomes of their work with Yang et al work.

Author Response

January 26, 2023

Prof. Dr. Samuel C. Mok

Editor-in-Chief, Cancers

Dear Editor:

Thank you very much for the review of our manuscript titled “The Efficacy and Safety of Immune Checkpoint Inhibitor and Tyrosine Kinase Inhibitor Combination Therapy for Advanced or Metastatic Renal Cell Carcinoma: A Multicenter Retrospective Real-world Cohort Study”.

We sincerely appreciate all valuable comments and suggestions, which helped us to improve the quality of our manuscript. Our responses to the reviewers’ comments are described below in a point-to-point manner. Appropriate changes, suggested by the Reviewers, have been introduced to the manuscript (track-changes mode in the red color font). Let me emphasize our full readiness to make any further improvements to the manuscript.

We hope that our manuscript will be acceptable for publication in the Cancers.

We look forward to hearing from you.

Yours sincerely,

Takuya Koie

Department of Urology

Gifu University Graduate School of Medicine

1-1 Yanagido, Gifu, Gifu 501-1194, Japan

TEL.: +81-582-30-6338

FAX: +81-582-30-6341

Responses to the reviewer's comments

We would like to thank the Reviewers for taking the time and effort necessary to review the manuscript. We sincerely appreciate all the valuable comments and suggestions, which helped us to improve the quality of the manuscript.

Response to Reviewer 2

The authors appreciate the reviewer’s comments. The authors’ point-by-point responses to the comments are given below.

  1. However, similar such study by Yang et al; has been already reported in “Combining immune checkpoint inhibition plus tyrosine kinase inhibition as first and subsequent treatments for metastatic renal cell carcinoma” published in Cancer Medicine 11, 2022, 3106.

The authors have not cited this paper. Authors need to discuss this paper in their manuscript and the need to compare the outcomes of their work with Yang et al work.

Response:

The authors have added the following references:

  1. Yang, Y.; Psutka, SP.; Parikh, AB.; Li, M.; Collier, K.; Miah, A.; Mori, SV.; Hinkley, M.; Tykodi, SS.; Hall, E.; et al. Combining immune checkpoint inhibition plus tyrosine kinase inhibition as first and subsequent treatments for metastatic renal cell carcinoma. Cancer Med. 2022, 11(16), 3106-3114.

The authors have added the following sentences in discussion on line 277:

As real-world data, Yang et al. reported the efficacy and safety of ICI+TKI in any treatment line for mRCC [28]. In this study, they found that 38.8% of patients received ICI+TKI as first line, with an ORR of 56.7% and a median PFS of 15.2 months [27].

Reviewer 3 Report

The manuscript "The Efficacy and Safety of Immune Checkpoint Inhibitor and Tyrosine Kinase Inhibitor Combination Therapy for Advanced or Metastatic Renal Cell Carcinoma: A Multicenter Retrospective Real-world Cohort Study" by Koie et al, submitted to Cancers, evaluated the efficacy and safety of combining immune checkpoint inhibitors (ICIs) and tyrosine kinase inhibitors (TKIs) in 51 patients with advanced or metastatic renal cell carcinoma (mRCC). The study found that patients aged 70 years and those with poor-risk mRCC had significantly poorer progression-free survival than their counterparts. Adverse events related to the ICI+TKI combination occurred in 84.3% of patients, with 43.1% experiencing grade 3 events.

The study is an outstanding addition to the clinical investigation on the use of ICI+TKI therapy in mRCC, but it should be noted that it is not a prospectively planned clinical trial and has some limitations. This work includes a mixture of patients, adverse events, stages of toxicity, and treatment regimens, and it is important for clinicians to know these data. there is a lack of translational data in the work, which is purely clinical observation. The cohort is relatively small and without strict control.

One suggestion for the authors to consider is to include a rationale for pooling cases with pure ICI or TKI treatment as controls in the Kaplan-Meier analysis. This could provide a better comparison of the efficacy and safety of the ICI+TKI combination therapy in mRCC.

Additionally, the high rate of hepatobiliary disorders in Grade 3/4 events should be double-checked and discussed. The authors should investigate if there is any specific reason for the high rate of these adverse events and provide more information on the management of these events.

Minor:

Please briefly introduce the target of drugs if first shown. e.g. sunitinib…

The format of Table 1 and Table 4 could be slightly changed for clarity, as the subtitle and content are not seen as straightforward.

Grammar and spell check is required, eg. capital of “Any Grade” in the summary and abstract. Moderate English changes are required.

Author Response

January 26, 2023

Prof. Dr. Samuel C. Mok

Editor-in-Chief, Cancers

Dear Editor:

Thank you very much for the review of our manuscript titled “The Efficacy and Safety of Immune Checkpoint Inhibitor and Tyrosine Kinase Inhibitor Combination Therapy for Advanced or Metastatic Renal Cell Carcinoma: A Multicenter Retrospective Real-world Cohort Study”.

We sincerely appreciate all valuable comments and suggestions, which helped us to improve the quality of our manuscript. Our responses to the reviewers’ comments are described below in a point-to-point manner. Appropriate changes, suggested by the Reviewers, have been introduced to the manuscript (track-changes mode in the red color font). Let me emphasize our full readiness to make any further improvements to the manuscript.

We hope that our manuscript will be acceptable for publication in the Cancers.

We look forward to hearing from you.

Yours sincerely,

Takuya Koie

Department of Urology

Gifu University Graduate School of Medicine

1-1 Yanagido, Gifu, Gifu 501-1194, Japan

TEL.: +81-582-30-6338

FAX: +81-582-30-6341

Responses to the reviewer's comments

We would like to thank the Reviewers for taking the time and effort necessary to review the manuscript. We sincerely appreciate all the valuable comments and suggestions, which helped us to improve the quality of the manuscript.

Response to Reviewer 3

The authors appreciate the reviewer’s comments. The authors’ point-by-point responses to the comments are given below.

  1. The study is an outstanding addition to the clinical investigation on the use of ICI+TKI therapy in mRCC, but it should be noted that it is not a prospectively planned clinical trial and has some limitations. This work includes a mixture of patients, adverse events, stages of toxicity, and treatment regimens, and it is important for clinicians to know these data. there is a lack of translational data in the work, which is purely clinical observation. The cohort is relatively small and without strict control.

Response:

The authors have added the following sentence on line 416:

Fourth, since this is an observational study, translational research is lacking, and this seems to be an issue for the future.

  1. One suggestion for the authors to consider is to include a rationale for pooling cases with pure ICI or TKI treatment as controls in the Kaplan-Meier analysis. This could provide a better comparison of the efficacy and safety of the ICI+TKI combination therapy in mRCC.

Response:

The authors have added their data to the discussion on line 282:

We also reported the efficacy and safety of NIVO + IPI for mRCC [10,11]. Namely, the ORR in moderate- and poor-risk patients based on the IMDC risk classification ranged from 34.3 to 41.9%, with OS and PFS at 6 months ranging from 94.7 to 100% and 76.1 to 78.6%, respectively, and OS and PFS at 12 months ranging from 89.1 to 95.8% and 56.2 to 63.1 %, respectively [10,11]. However, it is difficult to simply compare these studies because of the differences in patient background, time of investigation, and follow-up period. On the other hand, based on these results, it may be safe to state that OS and PFS are almost equivalent, although the ORR may be better with ICI+TKI than with NIVO+IPI.

The authors have added the following part on line 405:

First, because this is a retrospective, multicenter, single-arm study

  1. Additionally, the high rate of hepatobiliary disorders in Grade 3/4 events should be double-checked and discussed. The authors should investigate if there is any specific reason for the high rate of these adverse events and provide more information on the management of these events.

Response:

The authors have added the following sentences in discussion on line 387:

In phase III studies including JAVELIN RENAL 101, KEYNOTE 426, CheckMate 9ER, and CLEAR studies, ≥ grade 3 increases in alanine aminotransferase (ALT), aspartate aminotransferase (AST), ICI +TKI were observed in 3.1-7.0% and 3.4-13.3% of patients treated with TKIs [15-20,27]. In a subgroup analysis of Japanese patients in the JAVELIN RENAL 101 study reported by Uemura et al. [37], grade ≥3 ALT and AST elevations were found in 6.0% and 3.9% of the total population in the AVE+AXI group, compared with the incidence tended to be higher in the Japanese population, 12.1% and 6.0%, respectively, in the AVE+AXI group. According to a subgroup analysis of Japanese patients in the KEYNOTE 426 study, 11.4% of Japanese patients in the PEM+AXI group had liver dysfunction of grade 3 or higher [38]. I In the present study, the incidence of grade ≥3 hepatobiliary system disorders were 13.7%, which is comparable to the results of the subgroup analysis of Japanese patients. A higher incidence of TRAEs has been reported in Japanese or Asian patients with RCC who received VEGF/VEGF receptor inhibitors compared to non-Asian patients [39-42]. Therefore, it is possible that racial differences in TRAEs during treatment with ICI+TKIs may be observed, especially in Japanese patients, who may have more hepatobiliary diseases, and therefore, more careful follow-up may be necessary when treating mRCC with ICI+TKIs.

The authors have added the references:

  1. Uemura, M.; Tomita, Y.; Miyake, H.; Hatakeyama, S.; Kanayama, HO.; Numakura, K.; Takagi, T.; Kato, T.; Eto, M.; Obara, W.; et al. Avelumab plus axitinib vs sunitinib for advanced renal cell carcinoma: Japanese subgroup analysis from JAVELIN Renal 101. Cancer Sci. 2020, 111(3), 907-923.
  2. Tamada, S.; Kondoh, C.; Matsubara, N.; Mizuno, R.; Kimura, G.; Anai, S.; Tomita, Y.; Oyama, M.; Masumori, N.; Kojima, T.; et al. Pembrolizumab plus axitinib versus sunitinib in metastatic renal cell carcinoma: outcomes of Japanese patients en-rolled in the randomized, phase III, open-label KEYNOTE-426 study. Int J Clin Oncol. 2022, (1), 154-164.
  3. Ueda, T.; Uemura, H.; Tomita, Y.; Tsukamoto, T.; Kanayama, H.; Shinohara, N.; Tarazi, J.; Chen, C.; Kim, S.; Ozono, S.; et al. Efficacy and safety of axitinib versus sorafenib in metastatic renal cell carcinoma: subgroup analysis of Japanese patients from the global randomized Phase 3 AXIS trial. Jpn J Clin Oncol. 2013, 43(6), 616-28.
  4. Tomita, Y.; Fukasawa, S.; Oya, M.; Uemura, H.; Shinohara, N.; Habuchi, T.; Rini, BI.; Chen, Y.; Bair, AH.; Ozono, S.; et al. Key predictive factors for efficacy of axitinib in first-line metastatic renal cell carcinoma: subgroup analysis in Japanese patients from a randomized, double-blind phase II study. Jpn J Clin Oncol. 2016, 46(11), 1031-1041.
  5. Uemura, H.; Shinohara, N.; Yuasa, T.; Tomita, Y.; Fujimoto, H.; Niwakawa, M.; Mugiya, S.; Miki, T.; Nonomura, N.; Takahashi, M.; et al. A phase II study of sunitinib in Japanese patients with metastatic renal cell carcinoma: insights into the treatment, efficacy and safety. Jpn J Clin Oncol. 2010, 40(3), 194-202.
  6. Yoo, C.; Kim, JE.; Lee, JL.; Ahn, JH.; Lee, DH.; Lee, JS.; Na, S.; Kim, CS.; Hong, JH.; Hong, B.; et al. The efficacy and safety of sunitinib in Korean patients with advanced renal cell carcinoma: high incidence of toxicity leads to frequent dose reduction. Jpn J Clin Oncol. 2010, 40(10), 980-5.

  1. Please briefly introduce the target of drugs if first shown. e.g. sunitinib…

Response:

The authors have revised the following sentences on line 58:

Over the past decade, novel tyrosine kinase inhibitors (TKIs) targeting the VEGF receptor, including sunitinib (SUN), axitinib (AXI), and pazopanib, have emerged and have been used to treat advanced and metastatic RCC (mRCC) with completely different effects than previous therapies. Although sunitinib (SUN), axitinib (AXI), and pazopanib were once the standard of care for advanced and metastatic RCC mRCC, along with tyrosine kinase inhibitors (TKIs) targeting the VEGF receptor and mTOR inhibitors, dramatic changes in first-line therapy have occurred with the availability of immune checkpoint inhibitors (ICIs) [4].

  1. The format of Table 1 and Table 4 could be slightly changed for clarity, as the subtitle and content are not seen as straightforward.

Response:

The authors have revised the Table 1 and Table 4 and added the following sentence on line 171:

The number of patients treated with AVE+AXI, PEM+AXI, NIVO+CABO, and PEM+LEN were 19 (37.3%), 19 (37.3%), 8 (15.7%), and 5 (9.7%), respectively.

  1. Grammar and spell check is required, eg. capital of “Any Grade” in the summary and abstract. Moderate English changes are required.

Response:

The authors have confirmed and corrected Grammar and spell.

Round 2

Reviewer 2 Report

Authors have addressed the comments.